# Polariton design and modulation via van der Waals/doped semiconductor heterostructures

Mingze He [1,9], Joseph R. Matson [2,9], Mingyu Yu[3], Angela Cleri[4], Sai S. Sunku[5], Eli Janzen[6], Stefan Mastel[7], Thomas G. Folland [8], James H. Edgar [6], D. N. Basov [5], Jon-Paul Maria[4], Stephanie Law [3,4] & Joshua D. Caldwell [1,2] ✉

Hyperbolic phonon polaritons (HPhPs) can be supported in materials where the real parts of their permittivities along different directions are opposite in sign. HPhPs offer confinements of long-wavelength light to deeply sub-diffractional scales, while the evanescent field allows for interactions with substrates, enabling the tuning of HPhPs by altering the underlying materials. Yet, conventionally used noble metal and dielectric substrates restrict the tunability of this approach. To overcome this challenge, here we show that doped semiconductor substrates, e.g., InAs and CdO, enable a significant tuning effect and dynamic modulations. We elucidated HPhP tuning with the InAs plasma frequency in the near-field, with a maximum difference of 8.3 times. Moreover, the system can be dynamically modulated by photo-injecting carriers into the InAs substrate, leading to a wavevector change of ~20%. Overall, the demonstrated hBN/doped semiconductor platform offers significant improvements towards manipulating HPhPs, and potential for engineered and modulated polaritonic systems.

Due to the long free-space wavelength of mid- to far-infrared (IR) light, the realization of deeply subdiffractional photon confinement via the stimulation of polaritons[1] is critical for flat IR nanophotonic applications, such as miniaturized optical components[2,3], on-chip photonics, polariton waveguides[4] and nanolasers[5]. Specifically, hyperbolic polaritons supported in extremely anisotropic media, i.e., those featuring permittivity tensor components with opposite signs along different optical axes, can offer significant promise for many nanophotonic applications[6] where stronger confinement and improved control over propagation is beneficial. Applications of these properties include hyperlensing[7–9], metasurface-based optical components[10], quantum optics[11], and probes of nanoscale defects[12]. While hyperbolicity was first demonstrated with artificial dielectric/metal stacks[8], it was later discovered that a list of natural materials[1,13], including hexagonal boron nitride (hBN[14,15]), $MoO_3$[16,17], $V_2O_5$[18], support hyperbolic phonon polaritons (HPhPs). These opportunities are further expanded within low symmetry systems as demonstrated by the report of so-called 'Ghost-polaritons' in off-cuts of calcite[19] and hyperbolic shear polaritons in monoclinic crystals such as $\beta$-$Ga_2O_3$[20]. Such HPhPs in natural crystals feature exceptionally low optical losses[21–23], as the polaritons are derived from optic phonons[22,24] instead of scattering from free carriers[25].

Although HPhPs are volume-confined, they can still interact with the local environment through the evanescent field, and HPhP

[1]Department of Mechanical Engineering, Vanderbilt University, Nashville, TN 37240, USA. [2]Interdisciplinary Materials Science Program, Vanderbilt University, Nashville, TN 37240, USA. [3]Department of Materials Science and Engineering, University of Delaware, Newark, DE 19716, USA. [4]Department of Materials Science and Engineering, The Pennsylvania State University, University Park, Pennsylvania, PA 16802, USA. [5]Department of Physics, Columbia University, New York, NY 10027, USA. [6]Tim Taylor Department of Chemical Engineering, Kansas State University, Manhattan, KS 66506, USA. [7]Attocube Systems AG, Haar, Munich 8550, Germany. [8]Department of Physics and Astronomy, The University of Iowa, Iowa City, IA 52242, USA. [9]These authors contributed equally: Mingze He, Joseph R. Matson. ✉e-mail: josh.caldwell@vanderbilt.edu

wavevectors are demonstrated to be tuned and engineered by changing the substrate permittivity in a list of studies[4,26-31], with this effect having been generalized by ref. [26] On this track, one could use dynamic materials such as phase change materials[4,28,32] and graphene[33-38] to actively modulate the HPhPs supported in the heterostructure. Additionally, HPhPs propagating across domains with varying permittivities will be refracted, with the behavior described by Snell's law[26] and the continuity of tangential components of the wavevectors. With those fundamentals, patterned substrates can be used to manipulate polaritons supported in pristine hyperbolic media, such as polaritonic refraction[26,32,39] (e.g., prism and lensing), structured HPhPs[4,32,40] (e.g., waveguiding), photonic crystals[41,42] and accelerated HPhPs[43]. Like dielectric optics, those effects rely on wavevector differences for the HPhPs over different substrate regions, and some results can be enhanced with more significant contrast[43]. Additionally, the platform is ideally flat, as placing thin vdW materials over uneven surfaces (e.g., silicon pillars) will modify the morphology and/or induce strain[44] and scattering[45]. However, all existing demonstrations have either used 3D structures to induce a large contrast in the wavevectors, e.g., silicon versus air (etched silicon[40]), or provided fundamentally limited polaritonic wavevector change (-1.6 times for phase change materials[4,32] and -1.2–2 times for graphene/hyperbolic media[33-38]). Therefore, it is prudent to search for a platform that provides a planar surface and sufficient wavevector contrast that can be actively controlled.

Here, we demonstrate a hyperbolic material/doped semiconductor platform capable of controlling HPhPs with large tuning range with ultralow surface roughness (sub-nanometer). This platform allows for nearly continuous tuning of HPhP wavevectors, with maximum contrast of -8.3 times being experimentally demonstrated in the near-field. This doubles the value that can be achieved with noble metals and dielectrics. Moreover, we illustrate a sharp modal order transition when the plasma frequency of the doped semiconductor passes through the transitional frequency, leading to a wavevector discontinuity suitable for modulation and sensing applications. Finally, we show that the hBN/doped semiconductor system can be modulated by photo-injection, with an experimentally demonstrated polaritonic wavelength change of ~20% on a picosecond timescale. Although we focus primarily on uniformly doped semiconductors, we provided a proof-of-concept demonstration of hBN over in-plane varying plasma frequency with tuned wavevectors, offering significant freedom to manipulate HPhPs along a planar surface. Importantly, the platform is not limited to hBN, as plasma frequencies of doped semiconductors can be tuned over an extended range of frequencies for other hyperbolic materials (e.g., $\alpha$-$MoO_3$), providing a significant toolbox for manipulating HPhPs.

## Results

### Concept of tuning HPhPs via substrate permittivity

Although HPhPs are volume-confined modes, they remain sensitive to the local environment[4,26-31], e.g., the dielectric function of the substrate. The dependence of HPhP wavevectors ($k_{HPhP}$) over substrate permittivity can be described by an analytical solution[14]:

$$k(\omega) = k' + ik'' = -\frac{\psi}{d}\left[atan\left(\frac{\varepsilon_o}{\varepsilon_t \psi}\right) + atan\left(\frac{\varepsilon_s}{\varepsilon_t \psi}\right) + \pi l\right], \psi = -i\sqrt{\frac{\varepsilon_z}{\varepsilon_t}} \quad (1)$$

where $d$ represents the hBN thickness, $\varepsilon_o$ and $\varepsilon_s$ the complex dielectric functions of the superstrate (air here) and the substrate, respectively, and $\varepsilon_t$ and $\varepsilon_z$ are dielectric functions of hBN along the in and out of plane axes. $l$ is non-negative integer representing the HPhP mode order (0,1,2...), as infinite modes can be supported in hyperbolic systems simultaneously, and we here focus on the supported mode with lowest wavevector (referred to as fundamental mode[14,21]) because it usually is dominating in both near- and far-field studies. Note that the substrate-

induced wavevector difference is independent of hBN thickness, as Eq. (1) can be written in the following form:

$$k(\omega)d = -\psi\left[atan\left(\frac{\varepsilon_o}{\varepsilon_t \psi}\right) + atan\left(\frac{\varepsilon_s}{\varepsilon_t \psi}\right) + \pi l\right], \psi = -i\sqrt{\frac{\varepsilon_z}{\varepsilon_t}} \quad (2)$$

For a hBN/doped semiconductor heterostructure, the HPhPs supported can be indirectly engineered by changing the carrier concentration (therefore the plasma frequency, $\omega_p$, and dielectric function) of the underlying doped semiconductors, even with same hBN thickness. Therefore, HPhPs propagating in different domains possess different wavevectors, as shown in the schematic in Fig. 1a.

We first demonstrate the substrate-induced polariton tuning with a hBN/InAs heterostructure. To this end, we grew InAs samples with different $\omega_p$ by controlling the as-grown dopant concentration and transferred hBN slabs onto these InAs substrates. We then utilized scattering-type scanning near field microscopy (s-SNOM) to measure the HPhP dispersion via a standard procedure, and some exemplary data analyses are included in Supplementary Note 1. To minimize the role of hBN thickness in dictating HPhP wavevectors, similar hBN thicknesses (~51–55 nm) were used in this set of comparisons. The calculated dispersion plots along with the experimentally extracted data points validate that HPhP dispersions can be manipulated by varying the InAs $\omega_p$ (Fig. 1b–d). Notably, the HPhP modal number in those systems are different: $l = 0$ branch is only supported when InAs is dielectric (Fig. 1b, c versus Fig. 1d). Since the $l = 0$ branch is an even mode, the metallic substrate forbids the existence of this mode profile, with details greatly discussed in the ref. [27]. Thus, the $k_{HPhP}$ supported by uniform hBN can be engineered by the underlying doped semiconductor, for applications such as far-field resonators[4,26] or the near-field polariton propagation[4,26,32,39-41,43], so that etch-induced material damage to hBN can be avoided.

While it is possible to tune HPhPs by adjusting the substrate dielectric function, quantitative analysis of the dependence of the $k_{HPhP}$ upon the substrate permittivity is required enable the engineering of devices using this concept. Due to the substantial variation in hBN thickness between exfoliations, we discuss the influence of substrate permittivity ($\varepsilon_s$) on normalized wavevector $k(\omega)d$ in Eq. (2), as shown in Fig. 2. Notably, $l = 0$ branch is only supported when on a dielectric substrate, and it is forbidden over a metallic substrate due to the mirror symmetry[27]. Consequently, for the fundamental mode, the modal orders are different: $l = 0$ (low confinement) for dielectric substrates and $l = 1$ branch (high confinement) for metallic substrates. Notice that the cut-off of $l = 0$ branch happens at $Re(\varepsilon_s) = -\varepsilon_{superstrate}$ (air in our case) instead of $Re(\varepsilon_s) = 0$, and more related discussions are included in Supplementary Note 2. Furthermore, the wavevectors can also be manipulated by changing the substrate permittivity even within the same modal branch: increasing $Re(\varepsilon_s)$ leads to reduced $k_{HPhP}$. Importantly, the majority of $k_{HPhP}$ values are realized within a small range of $Re(\varepsilon_s)$ (between −10 and 10), encompassing both the highest and lowest $k_{HPhP}$ values and resulting in the largest polaritonic refraction effects.

While the tunability of HPhPs via substrate is promising, traditional metallic substrates (noble metals) do not provide access to the high wavevector range, as they exhibit $|Re(\varepsilon_s)|$ over 100 in the mid-infrared. Quantitatively, the maximum wavevector contrast achievable with noble metals and dielectrics (air in this comparison) is 4.2 times at 1500 cm$^{-1}$. To exploit the tunability of HPhP wavevectors, here we employed doped semiconductors, with the plasma frequencies tuned from below to above the Reststrahlen band of hBN. Therefore, we used InAs[47] and cadmium oxide (CdO[46]) as substrates, with achievable plasma frequencies being 500–2000 cm$^{-1}$ and 1500–15,000 cm$^{-1}$, respectively. Experimentally, we grew InAs and CdO substrates with varying $\omega_p$, and h$^{10}$BN flakes[21] were transferred onto them, with $k_{HPhP}$ measured and extracted by Fourier analysis (Supplementary Note 1).

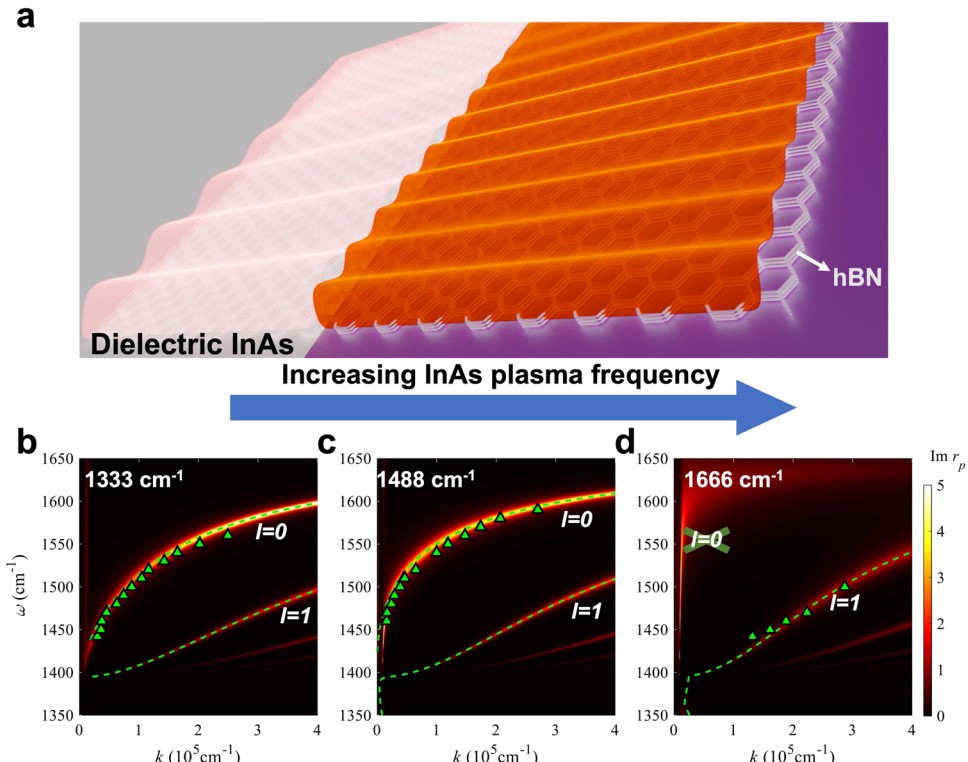

**Fig. 1 | Tuned hyperbolic phonon polariton (HPhP) dispersion of hexagonal boron nitride (hBN)/doped semiconductor heterostructure by controlling semiconductor plasma frequency ($\omega_p$). a** Schematic of the platform. For the same hBN, the HPhP wavelength changes as a function of the plasma frequency of the semiconductor. hBN is represented with multi-layer hexagonal structures, with HPhPs shown as waves over it. In this example, the InAs on the right side (magenta color) is metallic with sufficiently high carrier concentration, shrinking the HPhP wavelength. **b**–**d** Tuned HPhP dispersions by InAs plasma frequency. The plasma frequencies are noted on the corresponding panels, and we selected nearly identical thicknesses of hBN (51 nm, 51 nm and 55 nm respectively) to minimize the thickness dependence. The contour plots (imaginary part of near-field reflection, Im ($r_p$)) and dashed curves are calculated by transfer matrix method (TMM) and Eq. (1), respectively, and the triangles are extracted from s-SNOM data. The modal orders ($l$) are denoted on the corresponding HPhP branches.

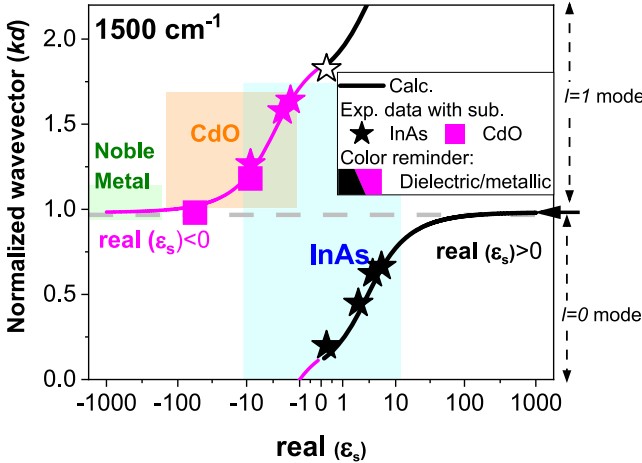

**Fig. 2 | Full control of HPhP wavevector ($k_{HPhP}$) with doped semiconductors.** The normalized $k_{HPhP}$ varies with the real part of the permittivity of the substrate (Re($\varepsilon_s$)). All curves are calculated by Eq. (2), while all symbols are experimental data. Colored shadows indicate the tunable range of $k_{HPhP}$ with that substrate with achievable doping. Solid symbols represent fundamental modes, while open symbols represent high-order modes. Data with dielectric (metallic) InAs are plotted with black (purple) stars. Data with metallic CdO are plotted with purple rectangles. Note that all noble metals lead to nearly identical $k_{HPhP}$, and the color box on the y-axis is extended for visualization purposes. Due to the accessible carrier concentration and high-frequency permittivity values, the tuning range of Re($\varepsilon_s$) of InAs and CdO are −10 to 10, and −100 to −1, respectively. All data above the horizontal dashed line are HPhPs with modal order above 0 ($l > 0$).

The experimental data are plotted as symbols in Fig. 2, showing excellent agreement with analytical solutions. Notably, with InAs substrates, we experimentally obtained both the highest and lowest $k_{HPhP}$ at 1500 cm$^{-1}$, with a $k_{HPhP}$ difference of ~8.3 times, which could be used to exploit polaritonic in-plane refractive behavior. We further compared different strategies of controlling HPhPs and concluded that our strategy provides the largest tunability and it could be universally applied to other hyperbolic systems (Supplementary Table 1).

With doped CdO and InAs, we have unlocked almost the entire potential of controlling the $k_{HPhP}$ through tuning substrate permittivity, with only a small range of $k_{HPhP}$ not accessible (~13%), where high refractive index dielectrics (|Re($\varepsilon_s$)| > 10) would be required. Unlike noble metals, which typically feature high surface roughness[47] (unless realized via specialized growth[47] and/or fabrication[45]), the doped semiconductors employed here possess low surface roughness (below 1 nm, Supplementary Note 3), which is crucial for HPhP platforms[45,47]. Importantly, semiconductors with varying in-plane $\omega_p$ can be realized (Supplementary Note 4), and we experimentally demonstrated that HPhPs supported in a hBN slab exhibit different wavevector in different domains, further facilitating the manipulation of HPhPs for applications like waveguiding[4], lensing[32] and resonators[4]. Therefore, those doped semiconductors can serve as ultrasmooth platforms to enable the tuning of HPhPs over an extended range of $k_{HPhP}$.

## Modal order transition in hBN/InAs heterostructures

To realize practical devices using doped semiconductor platforms, we must develop an understanding of the dependence of the wavevector on the substrate $\omega_p$. In particular, the hBN/InAs heterostructure offers an ideal tuning range for engineering the HPhPs. To that end, we

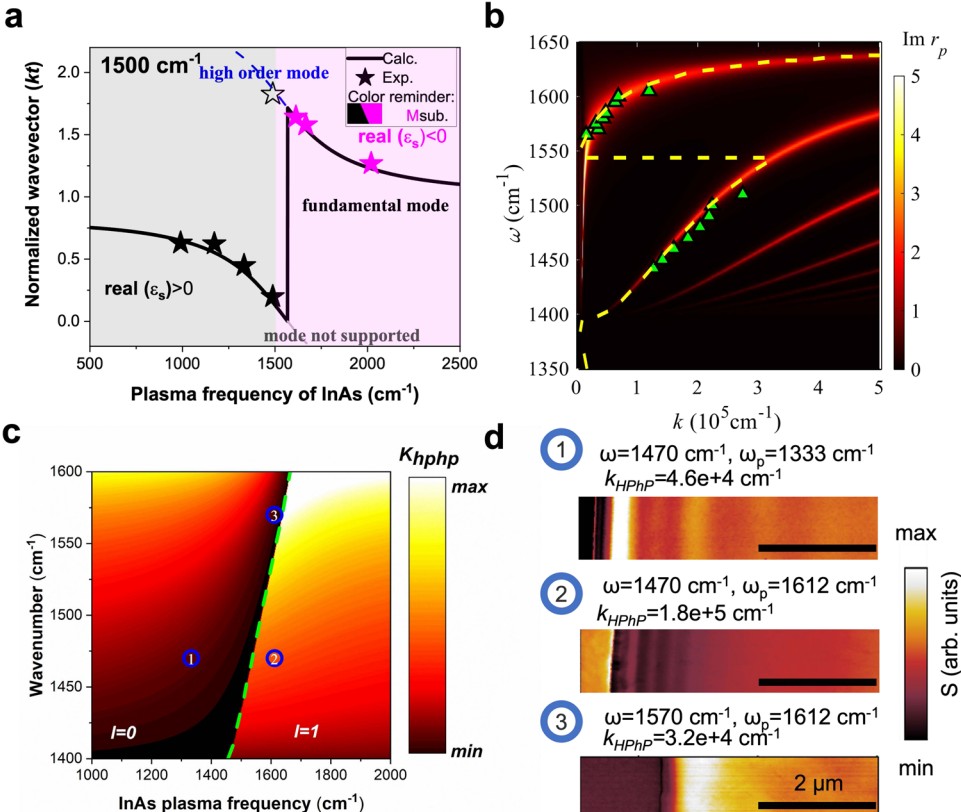

**Fig. 3 | Modal order transition of HPhPs in hBN/InAs heterostructure. a** The relationship between normalized $k_{HPhP}$ and InAs $\omega_p$. All curves are calculated by Eq. (1), while all symbols are experimental data. Solid symbols represent fundamental modes, while open symbols represent high-order modes. Data with dielectric (metallic) InAs are plotted with black (purple) symbols. The colored shades indicate whether InAs behaves as dielectrics or metals. **b** The modal order transition observed in the frequency domain. The contour plot is calculated by TMM and triangles are experimental data. For experimental data points above the transitional frequency, a different method is employed to extract wavevectors due to high polaritonic loss, and details are given in Supplementary Note 7. The yellow dashed curve is calculated with the analytical solution (Eq. (1)). **c** $k_{HPhP}$ at different

wavenumber and InAs $\omega_p$, with larger $k_{HPhP}$ plotted with the brighter color. The modal order clearly transits when InAs $\omega_p$ passes the Reststrahlen band of hBN, and the transition is noted with the green dashed curve. **d** Three representative s-SNOM images (optical amplitude, S) plotted with the same scale bar showing the engineered $k_{HPhP}$. The corresponding working frequency and InAs $\omega_p$ are noted as numbers in the (**c**). While the hBN thickness in subpanel−1 is 51 nm, the hBN thickness in subpanel-2,3 is 75 nm. The normalized wavevectors for the three subpanels are 0.235, 1.35, and 0.24, respectively. The nature of this tuning is the interaction between the evanescent field of HPhPs and the substrate, as indicated by the Eq. (2), and intuitively shown in Supplementary Fig. 18.

calculated how $k_{HPhP}$ changes with InAs $\omega_p$, at a working frequency of 1500 cm$^{-1}$, as shown in Fig. 3a. When the InAs carrier concentration increases such that the $\omega_p$ surpasses the excitation (working) frequency, InAs exhibit a change from a dielectric to metallic behavior (gray and magenta shaded in Fig. 3a, respectively). This dielectric to metallic evolution of the substrate further induces a HPhP modal order transition from $l=0$ to $l=1$ branch, leading to the wavevector discontinuity (Fig. 2). Additionally, in both dielectric and metallic regimes (gray and magenta shaded in Fig. 3a, respectively), an increase in InAs $\omega_p$ causes a decline in Re($\varepsilon_{InAs}$), resulting in a monotonical decrease in $k_{HPhP}$. By correlating $k_{HPhP}$ with InAs $\omega_p$, HPhP propagations can be manipulated with designable wavevectors over different domains, for both polaritonic refraction devices and/or resonators with significant design freedom. It is important to note that both minimum and maximum $k_{HPhP}$ occur around the transitional point, with 8.3 times difference being experimentally demonstrated.

Since the InAs permittivity is dispersive, the modal transition can also occur in the frequency domain. When the InAs Re($\varepsilon_s$) passes through −1 within the Reststrahlen band of hBN, a modal order transition occurs, splitting the fundamental HPhPs into two supported modal orders, as shown in Fig. 3b. As such, the dispersion plot features highly confined HPhPs below the transition frequency ($l=1$ mode), while showing reduced confinement of the HPhPs above it ($l=0$ mode)

(Fig. 3b). By using an InAs sample with a $\omega_p$ at a frequency within the Reststrahlen band, we experimentally demonstrated the transition discontinuity (Fig. 3b, green triangles), with good agreement with calculations. Note that the modal order transition (i.e., $\text{atan}\left(\frac{\varepsilon_o}{\varepsilon_t\psi}\right) + \text{atan}\left(\frac{\varepsilon_s(\omega_{tran})}{\varepsilon_t\psi}\right) = 0$) derived from Eq. (2) is thickness independent, yet it slightly varies with thickness when hBN is thicker than 150 nm, since the large wavevector assumption is invalid (Supplementary Note 5). The dispersion is analogous to two stacked hyperbolic dispersions, and we observed intriguing behaviors in both frequency (coexisting absorption and reflection modes) and real space (guiding in different regions at different frequencies) in numerical simulations (Supplementary Note 6).

The modal order transition criteria can be generalized for both the InAs $\omega_p$ (substrate-tuning) and modified working frequency, as shown in Fig. 3c. A clear transition line distinguishes the two modal orders, separating $k_{HPhP}$ into two regimes: one with highly confined HPhPs (right side) and one supporting less confined modes (left). At any transition point, the modal order transition could happen if the InAs $\omega_p$ or the working frequency is changed, i.e., along the InAs $\omega_p$ axis or wavenumber axis in Fig. 3c, respectively. Three representative s-SNOM images showing the highly and poorly confined HPhPs are presented in Fig. 3d, clearly showing the transitions. As a consequence of the modal order transition and wavevector discontinuity, around

the transition line, $k_{HPhP}$ experiences large variations with perturbations to the working conditions, e.g., InAs $\omega_p$, working frequency, and local environment (Supplementary Note 8). Therefore, a HPhP system working around the transition point, such as resonators, can be modulated effectively and/or used for refractive index sensing. It is important to note that high loss is associated with the large modulation depth around the transitional point, while it could be mitigated in certain applications (Supplementary Note 9). Note that for any working frequency inside the Reststrahlen band of hBN, there is a corresponding InAs $\omega_p$ to enable this transition, while such tunability of InAs[48] lends this concept to be expanded to other hyperbolic materials, e.g., MoO$_3$.

## Ultrafast modulation of HPhPs

Besides manipulating HPhPs in static structures, the hBN/doped semiconductor platform also enables the dynamic modulation of HPhPs. Both permittivity tensors of InAs and CdO can be modulated at the surface by electrical biasing[49] and photo-carrier injection[50,51], and here we demonstrate ultrafast modulation in a hBN/InAs heterostructure. When a ~100-fs, 80-MHz repetition rate pulsed laser source at 0.78 eV (1590 nm free-space wavelength) irradiates the hBN/InAs heterostructure, the photons will only be absorbed by InAs since it is above InAs bandgap (0.35 eV), yet well below that of hBN (5.95 eV)[52]. This process will generate a surplus of free-charge carriers at the InAs surface, locally increasing $\omega_p$ with a rise time below 1 ps, while those free carriers will subsequently recombine with a lifetime of ~8 ps, as shown in Fig. 4c. Because of this locally modulated InAs $\omega_p$, $k_{HPhP}$

values supported in the hBN/InAs heterostructure can be modulated at picosecond timescales.

To experimentally demonstrate such modulation, we conducted a near-field pump-probe measurement on one of the hBN/InAs heterostructures, with the detailed set-up provided in the Methods. Before the pump signal arrives (*pre-pump*), we observe a polariton wavelength of ~1.15 μm at 1450 cm$^{-1}$ for the HPhPs supported, with the InAs $\omega_p$ at 990 cm$^{-1}$. When the pump arrives (*at-pump*), the InAs $\omega_p$ at the surface is shifted to a higher frequency (1150 cm$^{-1}$), leading to a decreased $k_{HPhP}$ as mentioned, and we experimentally observe a stretched polariton wavelength by ~20% (~1.37 μm). Since the time constant of modulated InAs $\omega_p$ is ~8 ps, the modulated HPhP wavevector recovers within a similar temporal scale. Experimentally, the polaritonic behavior ($\lambda_{HPhP}$ = 1.29 μm) relaxes to a state between *pre-pump* and *at-pump* after 7 ps of the pump signal (green curve in Fig. 4a). Those effects are also manifested in the dispersion plots extracted from nano-FTIR scans (Supplementary Note 10). Importantly, the InAs $\omega_p$ was pumped from 990 cm$^{-1}$ to 1150 cm$^{-1}$, which are both well below the transition point discussed above. Therefore, we expect a stronger modulation if an InAs $\omega_p$ is modulated to surpass the transition frequencies and to induce modal order transitions. Note that in the ultrafast modulation, the only parameter changed in the system is the plasma frequency of InAs, i.e., the $\varepsilon_s$; thus, the modulation ratio ($\frac{k_{modulated}}{k_{static}}$) is independent of hBN thickness.

In addition to the modification of the polariton wavelength, we can also monitor how the resonant frequency of a given HPhP system change when the InAs is pumped. For a proof-of-concept

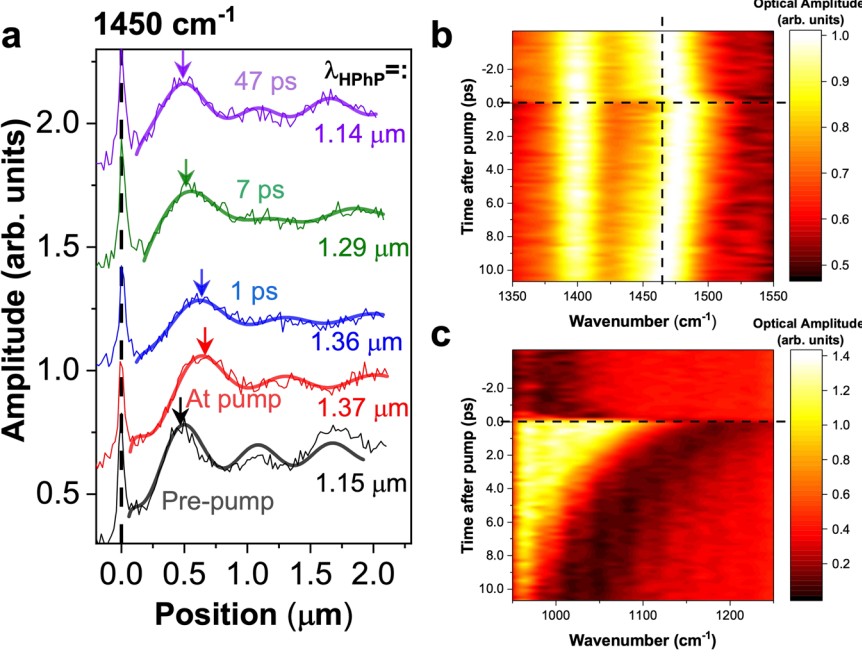

**Fig. 4 | Ultrafast modulation of HPhPs. a, The line profiles extracted from pump-probe nano-FTIR scans at different time delays.** The noisy curves are raw data, and we used a double-damped sine-wave function to extract the HPhP wavelengths (noted in the panel), with the fitted data plotted as thick curves, and the procedure was introduced in ref. 18. The error bars of all fittings are between 10 and 20 nm, which is significantly smaller than the polariton wavelength modulation (200 nm). The arrows indicate polariton wavelengths, showing a clear shifting. **b** The ultrafast nano-FTIR probed at a constant spatial position (~0.35 μm from the edge, $k_{HPhP} \approx 7 \times 10^4$ cm$^{-1}$). Since the distance from the edge (*dx*) does not correspond to $\lambda_p/2$, the $k_{HPhP}$ was determined by correlating the peak frequency with the dispersion plot. The dashed reference lines denote the static peak position and the zero-time delay, respectively. At this constant *dx*, the corresponding $k_{HPhP}$ is not expected to change as all the *dx* in (**a**) are ~0.45 times of polariton wavelength ($\lambda_p$).

The data here are processed with FFT filters to remove noise, and the raw data and signal process can be found in Supplementary Note 11. Another measurement at a different location shows a similar outcome (Supplementary Fig. 19). Note that the high optical amplitude at 1400 cm$^{-1}$ is the TO phonon of hBN, and it was not modulated in our configuration. **c** The ultrafast nano-FTIR measurements on InAs substrate. The reflection dip position is correlated to the plasma frequency of InAs, and the dielectric function fitting in the near-field can be found in Supplementary Fig. 14. The InAs static plasma frequency is 990 cm$^{-1}$, while the plasma frequency at pump is ~1150 cm$^{-1}$, and the hBN thickness is 55 nm. In the pump-probe setup, we approximate the InAs as uniformly pumped (valid for our system as discussed in Supplementary Note 12), and the transient InAs plasma frequencies were fitted with nano-FTIR spectra with a finite dipole model, and more details can be found in Supplementary Note 12.

demonstration, we measured the nano-FTIR spectra at a spatial position ~0.35 μm from the hBN edge, which corresponds to a constant wavevector of HPhP system ($k \approx 7 \times 10^4$ cm⁻¹) for tip launched mode, and the frequency amplitude peaks indicate the modal frequency. When InAs $\omega_p$ is increased, the wavevector at a fixed frequency is reduced; therefore, for a constant wavevector, the corresponding frequency blueshifts. Experimentally, the modal frequency indeed experienced a blue shift when the InAs was pumped, with a time constant of ~6 ps. This implies that we can modulate HPhP resonators at ultrafast time scales with un-pumped phonon materials for potentially lower loss, and the ultrafast switching of HPhPs has important implications for device applications in modulated optical sources[53], beacons and other areas. Importantly, we do not pump the polaritonic material itself (unlike reference where polar materials are directly pumped[54]), which can be a challenge due to wide bandgaps[52,55], and our approach can be universally applied to other HPhP supporting materials. Our approach thereby offers a great degree of flexibility in terms of doped substrates, as well as other hyperbolic materials, and could provide a foundation for more complex heterostructures.

## Discussion

In summary, we proposed and demonstrated a hyperbolic material/doped semiconductor platform to manipulate HPhPs with doubled tuning range compared with previous demonstrations. Our heterostructure platform offers significant improvements by providing access to almost all (~87%) possible HPhP wavevectors, with both maximum and minimum values accessible (~8.3 times difference experimentally demonstrated). This is in contrast to conventional noble metal and dielectric substrates that only provide access to discrete wavevectors with limited differences (~4.2 time). Moreover, when the plasma frequency of the doped semiconductor passes through the polariton frequency, a sharp modal order transition of HPhP will happen, and the HPhP system is sensitive to the local environment around the transition point, which could be used for sensing and modulation purposes. Finally, we demonstrate an ultrafast modulation of HPhPs at picosecond time scales by photoinjecting free carriers into semiconductors, with a polaritonic wavelength change of ~20%. Our approach demonstrates a highly feasible toolkit to control HPhPs with large tunability (Supplementary Table 1). Importantly, the rise time of the system (below 1 ps) is relatively short compared with the group velocity of HPhPs, which could potentially lead to time-varying effects as demonstrated in radiofrequencies[56]. Therefore, it could potentially be a platform to explore spatial-temporal effects in the real space, yet significant efforts are needed. With advances in semiconductor manufacturing, enabling further reduced ohmic losses and in-plane variations in doping[57], we expect doped semiconductors to be an increasingly important platform to manipulate HPhPs, in both the near- and far-fields. Importantly, the concept is not limited to hBN, and these effects can be realized over an extended spectral range and hyperbolic materials (e.g., α-MoO$_3$[16], β-Ga$_2$O$_3$[20], calcite[19]), opening a toolbox to manipulate in-plane HPhPs.

## Methods
### Device fabrication
In-doped CdO (n-type) was deposited on 2-inch r-plane (012) sapphire single crystal substates at 400 °C by a reactive co-sputtering process employing high-power impulse magnetron sputtering (HiPIMS) and radio frequency (RF) sputtering from 2-inch diameter metal cadmium and indium targets, respectively. HiPIMS drive conditions were 800-Hz frequency and 80-μs pulse time, yielding a 1250-μs period and 6.4% duty cycle. Film growth occurs in a mixed argon (20 sccm) and oxygen (14.4 sccm) environment at a total pressure of 10 mTorr. Post-deposition, samples were annealed in a static oxygen atmosphere at 635 °C for 30 min.

Si-doped InAs (n-type) was grown by molecular beam epitaxy (MBE) on the epi-ready GaAs (100) substrates, using a Veeco GENxplor

MBE in the University of Delaware Materials Growth Facility. In this system, the substrate temperature was measured by a band edge thermometer and the source flux was monitored as beam equivalent pressure (BEP). It provides an ultra-high vacuum environment with pressures as low as $1 \times 10^{-10}$ Torr for growth. GaAs substrates were fully deoxidized at 620 °C prior to growth. To prepare a smooth surface for the growth of Si-doped InAs, a GaAs buffer layer with a thickness of 100 nm was deposited at 580 °C first. Thereafter, the substrate was cooled to 420 °C. Then the Si-doped InAs layer was grown by opening the Si, In, As, and Bi source cells simultaneously. The As2:In BEP ratio and growth rate were kept around 20, and 1.7 um/h, respectively. A small amount of Bi (In:Bi BEP ratio was 50) was added as a surfactant to suppress the segregation of Si dopants on the surface[58]. The Si doping concentration was varied by changing the Si flux while using the same growth rate. The doping density and carrier mobility were detected by room-temperature Hall effect measurements in a van der Pauw configuration.

The doping concentrations in CdO and InAs were controlled during the growth process, and the carrier concentrations used in the Drude model (Eq. S1) were first estimated by room-temperature Hall effect measurements. Then we used FTIR to measure the reflection of those samples to fine tune the plasma frequency and scattering rate. Four representative FTIR fittings are shown in Supplementary Fig. 17.

$$\varepsilon_{(\omega)} = \varepsilon_s \left( 1 - \frac{\omega_p^2}{\omega^2 + i\omega\Gamma} \right) \qquad (3)$$

where $\varepsilon_s$ is the relatively permittivity of the undoped semiconductor, and $\omega_p$ is the plasma frequency, and $\Gamma$ is the scattering rate.

¹⁰B enriched hBN (~99% enriched[21,59]) flakes were exfoliated and transferred onto the InAs and CdO substrates using low contamination transfer techniques. The hBN crystals were grown with a boron source that was nearly 100% ¹⁰B isotope, as previously described[60].

### Calculations
In our heterostructures, the doped semiconductors are treated as substrate, as they are significantly thicker (~500 nm or 1 μm thick) than the evanescent field of HPhPs. The analytical solution is calculated by Eq. (1) in the main text, and the contour plots in Figs. 1, 3b are calculated by the transfer matrix method.

### Near-field measurements
Near-field nano-imaging experiments were carried out in a commercial Neaspec (www.neaspec.com) s-SNOM and nano-FTIR based around a tapping-mode atomic force microscope. A metal-coated Si-tip of apex radius $R \approx 20$ nm that oscillates at a frequency of $\Omega \approx 280$ kHz and tapping amplitude of about 70 nm is illuminated by a laser beam (probe laser, in the mid-infrared) at an angle 60° off normal to the sample surface. Scattered light launches HPhPs in the device and the tip then re-scatters light (described more completely in the main text) for detection in the far-field. Background signals are efficiently suppressed by demodulating the detector signal at harmonics of the tip oscillation frequency and employing pseudo-heterodyne interferometric detection.

**Static measurements.** For static measurements, the incident beam is from a single-frequency quantum cascade laser. The laser frequency can be tuned, and s-SNOM mapping were conducted in a single-wavelength measuring scheme. We extracted the second harmonic signal for the nano-FTIR data, while for s-SNOM we used the 3rd harmonic to attain more near-field information. The AFM tapping amplitude was ~100 nm pre-approach, while the tapping amplitude during measurement was ~60–80 nm. We further characterized the approach curve for tapping amplitude of 75 nm and 115 nm respectively, and the

3rd harmonic signals are both dominated by the near-field signal (Supplementary Fig. 20).

**Ultrafast measurements.** For ultrafast measurements, the probe laser is a pulsed broadband (~1000–2000 cm$^{-1}$) difference frequency generation laser (fiber laser), and the pulse width is ~200 fs after considering the dispersion of beam splitter. The pump laser is a pulsed single-frequency laser at 1590 nm, and the pulse width is ~100 fs. Those pulse widths set the time resolution of our system: ~0.3–0.4 ps. The repetition rates of both lasers are 80 MHz. The time delay is controlled by a delay stage on the pump beam line. In the ultrafast pumped heterostructure, we also notice that the HPhPs are experiencing a collective effect during the propagation in the temporal domain. HPhPs have small group velocities, and they will propagate for ~0.5–2 ps before being collected. As the lifetime of InAs free-carriers is ~8 ps (see fitting in Supplementary Note 12), the observed HPhPs will experience decreasing InAs $\omega_p$ during the propagation, i.e., increasing $k_{HPhP}$, if we assume uniform InAs pumping. Therefore, the as-measured ultra-fast HPhP response is convoluted within a certain time scale, depends on the group velocity.

## Data availability
Relevant data supporting the key findings of this study are available within the article and the Supplementary Information file. All raw data generated during the current study are available from the corresponding authors upon request

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

## Acknowledgements

M.H., J.R.M. and J.-P.M. gratefully acknowledge support for this work by Office of Naval Research Grant N00014-22-12035. J.-P.M. and J.D.C. acknowledge support from the Army Research Office Research Grant W911NF-21-1-0119. J.D.C. acknowledges support from the Office of Naval Research under the Twist-Optics Multi-University Research Initiative (MURI) N00014-23-1-2567. A.C. and J.-P.M. acknowledge support from the Army Research Office W911NF-16-1-0406. Support for hBN crystal growth was provided by the Office of Naval Research, award number N00014-22-1-2582. M.Y. and S.L. acknowledge funding from the National Science Foundation, Division of Materials Research under Award No. 1904760 and the Division of Electrical, Communications, and Cyber systems under Award No. 2102027. M.Y. and S.L. acknowledge the use of the Materials Growth Facility (MGF) at the University of Delaware, which is partially supported by UD-CHARM a National Science Foundation MRSEC under Award No. DMR-2011824. T.G.F. would like to acknowledge the University of Iowa startup funding. Work at Columbia was supported primarily by the Center on Precision-Assembled Quantum Materials, funded through the US National Science Foundation (NSF) Materials Research Science and Engineering Centers (award no. DMR-2011738). D.N.B. is the Vannevar Bush Faculty Fellow ONR-VB: N00014-19-1-2630.

## Author contributions

M.H., J.R.M., T.G.F., J.-P.M. and J.D.C. conceived the idea. M.H., S.S.S., T.G.F. and D.N.B. conducted the static near-field measurements, and M.H., J.R.M. and S.M. performed the ultrafast measurements. M.Y. and S.L. fabricated the InAs samples, and A.C. and J.-P.M. fabricated CdO samples. E.J. and J.H.E. grew the hBN crystal. M.H. carried out the modeling and data analysis. All participated in the writing.

## Competing interests

The authors declare no competing interests.
