## [Peer Review File · Nature Communications]

Polariton design and modulation via van der Waals/doped semiconductor heterostructureEditorial Note: Figures R1 and R2a in this Peer Review File have been amended to remove third-party material.

REVIEWER COMMENTS

Reviewer #1 (Remarks to the Author):

The manuscript titled "Polariton design and modulation via van der Waals / doped semiconductor heterostructures" by Joshua D. Caldwell presents an innovative approach to manipulating hyperbolic phonon polaritons (HPhPs) using doped semiconductors. The manuscript addresses the limitations of conventional noble metal and dielectric substrates in controlling the wavevector of HPhPs and proposes a novel platform that enables near-continuous tuning and access to a wide range of wavevectors. They demonstrate the efficacy of the hBN/doped semiconductor platform, showcasing its ability to achieve nearly continuous tuning and significantly improved wavevector contrast compared to conventional substrates. This introduces the concept of modal order transition and wavevector discontinuity, which are crucial for sensing and modulation applications. The authors illustrate how the plasmon frequency of the doped semiconductor plays a vital role in these phenomena, providing valuable insights for further exploration. The experimental results presented in the manuscript are impressive, particularly the demonstration of ultrafast modulation of HPhPs at picosecond timescales through carrier injection. This showcases the potential for dynamic control and manipulation of polaritonic systems.

Overall, the manuscript is well-written and provides valuable contributions to the field of nanophotonics. The findings have significant implications for the development of on-chip photonics and planar metasurface optics.

This manuscript will bring insight to a broader audience researching polaritons, 2D materials, and VdW materials beyond just hBN. This article is recommended for publication in Nature Communications after the following comments have been addressed.

Below are some comments I have:

1. In the abstract, consider rephrasing the sentence "This allows for greater control of polaritonic resonators and near-field polariton propagation without deleterious etching of hyperbolic materials" to improve clarity. Currently, non-destructive manipulation of hyperbolic materials for controlling hyperbolic polaritons has been achieved. Additionally, it would be helpful to provide a brief summary of both the experimental methodology and results in the abstract.
2. In the introduction, the authors did not mention the studies on the modulation of hyperbolic polaritons in heterostructures such as hBN/graphene (Nature Nanotechnology 2015, 10, 682) and MoO₃/graphene (ACS Photonics 2022, 9, 383. ; Nano Letters 2022, 22, 4260. ; Nature Communications 2022, 13, 3719. ; Science 2023, 379, 558. ; Nat Nano 2022, 17, 940). It is suggested that the authors include a table comparing different strategies for manipulating hyperbolic polaritons and plasmon polaritons. Additionally, the authors overlooked the classical work on structured hyperbolic phonon-polaritons (Nature Communications 2020, 11, 6086).
3. The article lacks simulation results of the field strength. It is highly necessary to include the simulated field distribution corresponding to Figure 3d. Simulation results would aid readers in understanding and making comparisons. If simulations cannot be provided, it should be mentioned.
4. Furthermore, without hBN covered, and the corresponding SNOM results for differently doped

semiconductor substrates under the same excitation conditions should be presented.

5. How was the doping level determined? The authors should provide a detailed interpretation.

6. The quality of irradiated CdO in Figure S4 seems to have suffered. The formation of many grain boundaries may affect the effectiveness of plasmons. The authors should discuss this aspect.

7. The thickness dependence of phonon polaritons in van der Waals materials, such as hBN and α -MoO₃, is of fundamental importance. It is also crucial to study how the thickness of hBN affects the modal order transition in hBN/InAs heterostructures.

8. Similarly, the impact of different hBN thickness and excitation wavelengths on the ultrafast modulation of hyperbolic phonon polaritons is an intriguing aspect for readers.

9. The quality of line profiles extracted from pump-probe nano-FTIR scans at different time delays is very poor. Furthermore, the authors visually estimated the wavelengths, which is not a valid procedure in Figure 4a. The standard procedure is to fit a damped oscillatory function with a wavelength parameter. The authors should fit a function to the profiles and extract the polariton wavelengths with error bars (95% confidence intervals).

Reviewer #2 (Remarks to the Author):

In this manuscript, the authors experimentally demonstrated that the wavelength of phonon polaritons residing in the hBN film can be controlled by engineering the plasma frequency (essentially the permittivity) of the semiconductor material (CdO and InAs) working as supported substrates. As a result, a broad range of polariton wavelength can be accessed. Furthermore, an ultrafast modulation of the polariton wavelength has been shown by using an optical pumping pulse to modify the transient permittivity of InAs, which is interesting. Here are several points that I believe important to address:

(1) Manipulating the wavelength of the phonon polariton supported in a MoO₃ film by actively tuning the conductivity of a nearby graphene sheet has been previously explored in Nat Commun 13, 3719 (2022), Nat. Nanotechnol. 17, 940–946 (2022), ACS Photonics 9, 383 (2022), and Science 379, 558 (2023). In this regard, the truly novel and interesting results of current manuscript are in its last part: modulating the polariton wavelength by ultrafast optical pumping. This type of ultrafast modulation has been previously proposed in theory [PRL 125, 037403 (2020)]. The authors may want to further clarify the novelty of their current work, and emphasize more on the last part of their study.

(2) In those applications mentioned in the introduction part of current manuscript, many of them are requiring high-quality polaritons. The authors may want to show and discuss quality factor of the polariton modes as the substrate property is changing. In case the losses are high in the experiment, the authors may want to reargue the potential applications of their current platform.

(3) The authors may want to discuss the detailed method to determine the transient plasma frequency of the substrate material after optical pumping in the experiment.

This work might be publishable after my above concerns are properly addressed.

Reviewer #3 (Remarks to the Author):

Review for “Polariton design and modulation via van der Waals / doped semiconductor heterostructures” by Mingze He et al. NCOMM-434944

In the manuscript, the authors study the tunability and manipulation of hyperbolic phonon polaritons using hyperbolic material/doped semiconductor platform. The study explores such tuning by utilizing the plasma frequency of an InAs/CdO substrate for nearly continuous HPhP wavevector tuning. Also, the authors show a sharp modal order transition when the plasma frequency of the doped semiconductor passes through the transitional frequency. Last, using ultrafast pulses, they show active modulation at picosecond timescales by photo-injecting carriers into the InAs substrate, showcasing a dynamic wavevector change of approximately 20%.

The authors have performed thorough research on the influence of substrate doping on the tunability of HPhP, combining theoretical and experimental research activities. I find the manuscript nicely written and scientifically interesting. Thus, in general, I can support publication in Nat. Comm. However, I think that the manuscript still lacks several important information and needs further clarifications, thus, I cannot recommend publication in its current appearance.

The issues are described below (non-prioritized):

- I find the ultrafast response of the system very interesting. Not many groups can perform time-resolved scanning near-field optical microscopy and retrieve transient nano-imaging capabilities. However, I feel that the presented results and the discussion are quite laconic. For example:
 - o The authors show the response only at a specific spatial position from the edge (position $\sim 0.35 \mu\text{m}$ from the hBN edge). Have they checked other locations? Is the temporal dynamics remain the same in each location? I guess there is a spatio-temporal dynamics, but it could be a loose coupling in this system. I think that at least another measurement in a different location or two will check/resolve this issue.
 - o Also, for different sample geometries and boundaries, the reflection from the boundaries suppose to vary, and in some cases multiple reflections from edges can be accumulated. Can/should this influence the temporal dynamics? for sure it will cause interferences. I think some discussion is missing.
 - o From the ultrafast dynamic evolution viewpoint – the pump causes free electron evolution, which is observed by the probe in various wavelengths. Have the authors examined what is the influence on the pump’s intensity?
 - o Any information on the group velocity of the phonon polaritons via such measurements?

- The authors mention that other hyperbolic materials, such as MoO₃, can be an alternative to hBN. Still, it seems to be that interesting research works on tunable phonon-polaritons with MoO₃ relevant to the

manuscript were not mentioned:

o Ruta, Francesco L., et al. "Surface plasmons induce topological transition in graphene/ α -MoO₃ heterostructures." *Nature Communications* 13.1 (2022): 3719.

o Zhang, Qing, et al. "Hybridized hyperbolic surface phonon polaritons at α -MoO₃ and polar dielectric interfaces." *Nano Letters* 21.7 (2021): 3112-3119.

- Some technical clarifications:

- o The demodulation order for the s-SNOM measurement is not specified in the paper or the supplementary materials. Specifying the order of the demodulation will provide information on the degree of near-field vs. far-field response.

- o The measurements are done with AFM's tip modulation of 100nm (for wavelengths of \sim 6 μ m to 10 μ m). This condition can cause the addition far-field information in the measured data. It will be nice to see the approach curve of the tips to observe the amount of far-field contribution.

- How repeatable are the measurements for different samples? The author reported the results in several thicknesses. Do they have two samples with the same thicknesses? How close are the measured results?

- Clarifications on Figures:

- o Figure 1a – could be more informative. Very nice illustration, but laconic. More graphical information of the various parameters (wavevector, thickness, etc.) can be added. The transition in Fig 1. B and C are both with thicknesses of 51nm (see caption)? If yes, why those are with different dispersion curves.

- o Figure 2 – a proper legend is missing. Most of the caption text is devoted to explaining the data. I think adding a legend with information on the different symbols will help. (metallic CdO - purple rectangles, etc.). Same comment for Figure 3a.

To conclude, I find the manuscript scientifically interesting and informative, yet I feel that it still lacks some information, thus I cannot recommend publication in its current appearance. With these points adequately addressed, the manuscript will likely merit being published in *Nat. Comm.*

Response letter

REVIEWER

COMMENTS

Reviewer #1 (Remarks to the Author):

The manuscript titled "Polariton design and modulation via van der Waals / doped semiconductor heterostructures" by Joshua D. Caldwell presents an innovative approach to manipulating hyperbolic phonon polaritons (HPhPs) using doped semiconductors. The manuscript addresses the limitations of conventional noble metal and dielectric substrates in controlling the wavevector of HPhPs and proposes a novel platform that enables near-continuous tuning and access to a wide range of wavevectors. They demonstrate the efficacy of the hBN/doped semiconductor platform, showcasing its ability to achieve nearly continuous tuning and significantly improved wavevector contrast compared to conventional substrates. This introduces the concept of modal order transition and wavevector discontinuity, which are crucial for sensing and modulation applications. The authors illustrate how the plasmon frequency of the doped semiconductor plays a vital role in these phenomena, providing valuable insights for further exploration. The experimental results presented in the manuscript are impressive, particularly the demonstration of ultrafast modulation of HPhPs at picosecond timescales through carrier injection. This showcases the potential for dynamic control and manipulation of polaritonic systems. Overall, the manuscript is well-written and provides valuable contributions to the field of nanophotonics. The findings have significant implications for the development of on-chip photonics and planar metasurface optics.

This manuscript will bring insight to a broader audience researching polaritons, 2D materials, and VdW materials beyond just hBN. This article is recommended for publication in Nature Communications after the following comments have been addressed.

We appreciate the reviewer's suggestions and efforts. Please find the point-to-point changes below.

Below are some comments I have:

1. In the abstract, consider rephrasing the sentence "This allows for greater control of polaritonic resonators and near-field polariton propagation without deleterious etching of hyperbolic materials" to improve clarity. Currently, non-destructive manipulation of hyperbolic materials for controlling hyperbolic polaritons has been achieved. Additionally, it would be helpful to provide a brief summary of both the experimental methodology and results in the abstract.

We appreciate the suggestions. We have revised the abstract accordingly:

Compared to surface polaritons, HPhPs offer further confinement of long-wavelength light to deeply subdiffractional scales. Additionally, the evanescent field of volume-confined propagation allows for interactions with substrates, enabling the tuning of HPhPs by altering the underlying materials. Therefore, control of polaritonic resonators and near-field polariton propagation can be realized with intact hyperbolic media without deleterious etching.

Also:

To overcome this challenge, we utilized doped semiconductors, e.g., InAs and CdO, for near-continuous tuning and access to both the maximum and minimum wavevectors, and we experimentally demonstrated ~8.3 times wavevector difference in the near-field.

2. In the introduction, the authors did not mention the studies on the modulation of hyperbolic polaritons in heterostructures such as hBN/graphene (Nature Nanotechnology 2015, 10, 682) and MoO₃/graphene (ACS Photonics 2022, 9, 383. ; Nano Letters 2022, 22, 4260. ; Nature Communications 2022, 13, 3719. ; Science 2023, 379, 558. ; Nat Nano 2022, 17, 940). It is suggested that the authors include a table comparing different strategies for manipulating hyperbolic polaritons and plasmon polaritons. Additionally, the authors overlooked the classical work on structured hyperbolic phonon-polaritons (Nature Communications 2020, 11, 6086).

We apologize for missing those references. We have revised the main text to highlight those related works:

Although HPhPs are volume-confined, they can still interact with the local environment through the evanescent field, and HPhP wavevectors are demonstrated to be tuned and engineered by changing the substrate permittivity in a list of studies¹⁻⁷, with this effect having been generalized by Fali. et al². On this track, one could use dynamic materials such as phase change materials⁸⁻¹⁰ and graphene¹¹⁻¹⁶ to actively modulate the HPhPs supported in the heterostructure.

And:

However, all existing demonstrations have either used 3D structures to induce a large contrast in the wavevectors, e.g., silicon versus air (etched silicon⁴³), or provided fundamentally limited polaritonic wavevector change (~1.6 times for phase change materials^{33,42} and ~ 1.2-2 times for graphene/hyperbolic media³⁵⁻⁴⁰).

And a table in the SI:

Table S1. Strategies of tuning and modulating HPhPs

Tuning strategy	Reference	Tuning range	Speed if dynamic
Structuring of hyperbolic media	^{19,20}	N.A. (Highlight structured HPhP instead of wavevector control)	Static
Substrate tuning with phase change	⁸⁻¹⁰	~60%	Nanoseconds
Substrate tuning with graphene	¹¹⁻¹⁶	~20% experimental ¹¹ and ~100% simulation	Not characterized. Could be picosecond level
Tuning with static substrate	¹⁻⁷	~400%	Static
Substrate tuning with static doped semiconductor	This work	~800%	Static
Optically pumped doped semiconductor	This work	~20%	Picoseconds

3. The article lacks simulation results of the field strength. It is highly necessary to include the simulated field distribution corresponding to Figure 3d. Simulation results would aid readers in understanding and making comparisons. If simulations cannot be provided, it should be mentioned.

We appreciate the reviewer for pointing it out. We have added the data to the supplementary and refer to it in the main text.

Main text:

The nature of this tuning is the interaction between the evanescent field of HPhPs and the substrate, as indicated by the equation (2), and intuitively shown in Fig. S18.

SI:

Fig. S18. The cross-sectional field profiles of hBN/InAs heterostructures. The simulation parameters (frequency, thickness and InAs ω_p) are identical to Fig. 3d. The polaritonic wavelength tuning is through the interaction between the evanescent field of HPhPs and the substrate.

4. Furthermore, without hBN covered, and the corresponding SNOM results for differently doped semiconductor substrates under the same excitation conditions should be presented.

We appreciate the reviewer's request. We have included the pump-probe data in the frequency range outside of the Reststrahlen band of hBN in Fig. 4c, and the spectral features are caused by the modulated InAs permittivity. We also performed s-SNOM measurements on InAs regions but we could not see the propagating SPPs supported by InAs. We attribute it to the low polariton lifetimes of SPPs (~0.1-0.5 ps).

5. How was the doping level determined? The authors should provide a detailed interpretation.

We have added more details about the doping level measurements. In short, the carrier concentrations and scattering rates of InAs and CdO samples were initially estimated by room-temperature Hall effect, and then those values were fine tuned by fitting the FTIR reflection measurements. The following discussions were added to the supplementary information.

The doping concentrations in CdO and InAs were controlled during the growth process, and the carrier concentrations used in the Drude model (Eq. S1) were first estimated by room-temperature Hall effect measurements. Then we used FTIR to measure the reflection of those samples to fine tune the plasma frequency and scattering rate. Four representative FTIR fittings are shown in Fig. S17.

$$\varepsilon(\omega) = \varepsilon_s \left(1 - \frac{\omega_p^2}{\omega^2 + i\omega\Gamma}\right) \quad \text{Eq. S1}$$

where ε_s is the relatively permittivity of the undoped semiconductor, and ω_p is the plasma frequency, and Γ is the scattering rate.

Figure S17. The fitting of FTIR data to determine plasma frequencies of InAs.

6. The quality of irradiated CdO in Figure S4 seems to have suffered. The formation of many grain boundaries may affect the effectiveness of plasmons. The authors should discuss this aspect.

We apologize that we failed to stress more clearly that the surface roughness was caused by photoresist residue instead of intrinsic CdO property, which can be optimized in the future. We have modified the caption and the text to more explicitly state it.

(a) S-SNOM image of hBN over in-plane irradiated CdO at 1442 cm^{-1} . The two lines denote where line profiles are taken to extract wavevectors. Note that the high surface roughness here ($\sim 5 \text{ nm}$) is caused by photoresist residues, which increases polariton scattering. We expect the surface roughness can be reduced to $\sim 1 \text{ nm}$ upon further procedure optimization, as the surface roughness of pristine CdO is $\sim 0.5 \text{ nm}$ (SI, section S3).

7. The thickness dependence of phonon polaritons in van der Waals materials, such as hBN and $\alpha\text{-MoO}_3$, is of fundamental importance. It is also crucial to study how the thickness of hBN affects the modal order transition in hBN/InAs heterostructures.

We appreciate the reviewer for pointing this out. One of the motivations of this study is to control the polariton characteristics with an intact van der Waals material (i.e., no thickness variations). Thus, we normalized the thickness dependence because those effects can be applied to hBN of any thicknesses. Since the thickness dependence has been extensively studied in references^{21,22}, we did not include related data and discussions in the original submission. For the thoroughness of this study, we further discussed the thickness dependence of the modal order transition.

In short, the transition frequency slightly shifts to higher frequency with thicker hBN. In the context of hBN thinner than 150 nm , the transition is barely dependent on the layer thickness. We have added a section in the supplementary information:

Here we discuss the thickness dependence of the modal order transition. Based on the analytical solution (assuming $k_{\text{HPhPs}} \gg k_0$), the modal order transition should not be thickness dependent. The analytical solution for the HPhP wavevector is:

$$k(\omega) = k' + ik'' = -\frac{\psi}{d} \left[\text{atan} \left(\frac{\varepsilon_o}{\varepsilon_t \psi} \right) + \text{atan} \left(\frac{\varepsilon_s(\omega)}{\varepsilon_t \psi} \right) + \pi l \right], \quad \psi = -i \sqrt{\frac{\varepsilon_z}{\varepsilon_t}} \quad \text{Eq. S1}$$

which can be rewritten as:

$$k(\omega)d = -\psi \left[\text{atan} \left(\frac{\varepsilon_o}{\varepsilon_t \psi} \right) + \text{atan} \left(\frac{\varepsilon_s(\omega)}{\varepsilon_t \psi} \right) + \pi l \right], \quad \psi = -i \sqrt{\frac{\varepsilon_z}{\varepsilon_t}} \quad \text{Eq. S2}$$

and the transition happens for hBN of any thickness when

$$\text{atan} \left(\frac{\varepsilon_o}{\varepsilon_t \psi} \right) + \text{atan} \left(\frac{\varepsilon_s(\omega_{\text{tran}})}{\varepsilon_t \psi} \right) = 0 \quad \text{Eq. S3}$$

where l is a non-negative integer representing the HPhP mode order (0,1,2,...). d represents the hBN thickness, ε_o and ε_s the complex dielectric functions of the superstrate (air here) and the substrate, respectively, and ε_t and ε_z are dielectric functions of hBN along the in and out of plane axes. The 0-order mode is no longer supported when $\text{atan} \left(\frac{\varepsilon_o}{\varepsilon_t \psi} \right) + \text{atan} \left(\frac{\varepsilon_s}{\varepsilon_t \psi} \right)$ is negative, which is referred to as modal order transition in the main text. We note that as the wavevector is both frequency and thickness dependent in hyperbolic media, solving Eq. S2 for the $k(\omega)d$ product normalizes the thickness dependence of the wavevector and thus, generalizes the solution to arbitrary thicknesses of hBN within the thin film limit (see further discussions on this limit below).

For thin hBN, the above discussion is precise as Eq. S2 is accurate, since the HPhP is very dispersive and the wavevectors are generally much larger than k_0 , as validated by experiments (**Fig. S6**). However, for thick hBN, the wavevectors are reduced and the accuracy of the analytical solution degrades in a wider frequency range, affecting the modal order transition criteria. For thick hBN, e.g., 300 nm, at a frequency that is higher than ω_{tran} in Eq. S4, the k_{HPhP} becomes large enough to satisfy the large wavevector assumption. Therefore, the transition frequency is higher for thick hBN, as shown in **Fig. S5**.

Figure S5. The dispersion plot of hBN/InAs ($\omega_p=1612 \text{ cm}^{-1}$) with different thicknesses. The InAs ω_p is 1612 cm^{-1} . The reference lines denote the range of transition frequencies with different hBN thicknesses. Note that the transition frequency does not change much with hBN thinner than 150 nm.

Figure S6. The dispersion plot of hBN/InAs with different hBN thicknesses (labeled in the figure). The InAs ω_p is 1612 cm^{-1} in all three plots (also same as **Fig. S5**), validating the calculations. The experimental data are plotted as green triangles and the analytical solution is plotted as dashed curves.

We also revised the main text to highlight the thickness dependence of the modal order transition.

Note that the modal order transition (i.e., $\text{atan}\left(\frac{\epsilon_o}{\epsilon_t \psi}\right) + \text{atan}\left(\frac{\epsilon_s(\omega_{tran})}{\epsilon_t \psi}\right) = 0$) derived from Eq. (2) is thickness independent, yet it slightly varies with thickness when hBN is thicker than 150 nm, since the large wavevector assumption is invalid (SI, section 5).

8. Similarly, the impact of different hBN thickness and excitation wavelengths on the ultrafast modulation of hyperbolic phonon polaritons is an intriguing aspect for readers.

We appreciate the reviewer for pointing this out. For the excitation wavelength, we are limited to 1560 nm because that is the only ultrafast excitation laser we have available. Since any excitation above the bandgap will induce free-carriers, which in turn will lead to the ultrafast modulation of HPhPs, we expect similar results with other excitation frequencies.

The modulation ratio, i.e., $\frac{k_{modulated}}{k_{static}}$, is thickness independent as discussed in the previous comment: only term ϵ_S is changed, which is the same mechanism of static state tuning. We added related discussions in the main text:

In the ultrafast modulation, the only parameter changed in the system is the plasma frequency of InAs, i.e., the ϵ_S ; thus, the modulation ratio ($\frac{k_{modulated}}{k_{static}}$) is independent of hBN thickness.

Note that the above discussion is assuming that the substrate has uniform carrier concentration, which is true for static semiconductor in this study. For pump-probe measurement, we added more discussions to justify our assumption that the InAs is uniformly pumped in both lateral and depth dimensions.

All the above discussions assume a uniform InAs pump, and our system can be approximated in such a way.

We first discuss the beam profile in the x-y plane. The pump laser (1560 nm wavelength) is focused by an off-axis parabolic mirror (NA=0.7). Because of the limited beam diameter of the pump beam (~2-3 mm) and much larger focal length (larger than 10 mm), the focal spot is relatively large (at least 3-5 λ , i.e., 5-8 μ m). Due to the challenge of the alignment, the actual focal spot is even larger than the calculated optimal condition. In this case, the power variation within the collected region (~2 μ m) is below 10% and can be neglected, and the InAs can be approximately considered uniform pumped.

We then discuss the absorbed energy at different depths, since the absorbed energy (i.e., the excited carrier concentration) decays when the pump laser penetrates into the InAs material. For this purpose, we calculated the absorption at different depths using a layer-resolved absorption calculation²³, and the absorption at 100 nm below the surface is only decreased by ~8% as compared to the top surface. As both nano-FTIR measurements in Fig. 4c and HPhP are dominated by the ~100 nm InAs below the surface^{24,25}, the carrier concentration variation in the z-axis can be neglected.

9. The quality of line profiles extracted from pump-probe nano-FTIR scans at different time delays is very poor. Furthermore, the authors visually estimated the wavelengths, which is not a valid procedure in Figure 4a. The standard procedure is to fit a damped oscillatory function with a wavelength parameter. The authors should fit a function to the profiles and extract the polariton wavelengths with error bars (95% confidence intervals).

We apologize that the procedure was not clearly described. We have modified the caption to more explicitly describe the fitting:

The noisy curves are raw data, and we used a double-damped sine-wave function to extract the HPhP wavelengths (noted in the panel), with the fitted data plotted as thick curves, and the procedure was

introduced in reference ²⁶. The error bars of all fittings are between 10 and 20 nm, which is significantly smaller than the polariton wavelength modulation (200 nm).

The data quality of nano-FTIR is normally worse than single-frequency s-SNOM measurements mostly for two reasons: (1) the laser power of the broadband laser (~0.2-0.5 mW) is much lower than a single-frequency quantum-cascade laser (~1-3 mW); (2) the spectral resolution is ~6 cm⁻¹ causing a certain level of convolutions. In a reference highlighting the hyperspectral imaging with nano-FTIR²⁷, the signals acquired with nano-FTIR were also significantly worse than the s-SNOM images. In our case, we also have time limitations in the measurements: we must perform the same line scan at different time delays, and we thus have stringent limitations on how much we can increase the integration time to improve the signal. For instance, each linescan as presented took approximately 1-2 hours to collect. We note that as the signal to noise ratio scales as the square root of the number of scans in the signal average, even a two fold increase in signal to noise would require a quadrupling of the measurement time, which is not possible. As such, as much as we wish this could be improved, given current lab constraints this is the best data that can currently be obtained within the operating conditions available.

[REDACTED]

Fig. R1 [REDACTED]. Data quality comparison between nano-FTIR and s-SNOM imaging. The data are taken from reference ²⁷.

Reviewer #2 (Remarks to the Author):

In this manuscript, the authors experimentally demonstrated that the wavelength of phonon polaritons residing in the hBN film can be controlled by engineering the plasma frequency (essentially the permittivity) of the semiconductor material (CdO and InAs) working as supported substrates. As a result, a broad range of polariton wavelength can be accessed. Furthermore, an ultrafast modulation of the polariton wavelength has been shown by using an optical pumping pulse to modify the transient permittivity of InAs, which is interesting. Here are several points that I believe important to address:

We appreciate the effort and suggestions from the reviewer, and please find our modifications and response below.

(1) Manipulating the wavelength of the phonon polariton supported in a MoO₃ film by actively tuning the conductivity of a nearby graphene sheet has been previously explored in Nat Commun 13, 3719 (2022), Nat. Nanotechnol. 17, 940–946 (2022), ACS Photonics 9, 383 (2022), and Science 379, 558 (2023). In this regard, the truly novel and interesting results of current manuscript are in its last part: modulating the polariton wavelength by ultrafast optical pumping. This type of ultrafast modulation has been previously

proposed in theory [PRL 125, 037403 (2020)]. The authors may want to further clarify the novelty of their current work and emphasize more on the last part of their study.

While we agree that tuning HPhPs with SPP supporting materials has been demonstrated to a great extent, our note that our work does not rely on the hybridization of HPhPs and SPPs as in most of those efforts, while also providing a significantly improved tuning range (~8 times) that is not accessible with graphene (~1.2 to 2 times). In the demonstration of coupling HPhPs with graphene SPPs by Dai. et al¹¹, the polariton wavevector is changed by adjusting the Fermi energy of graphene by ~20%. In mentioned references¹² and²⁸, the HPhPs wavevector can be tuned by ~60-100%. This is due to the small interacting volume between HPhP fields and graphene. In the highlighted theoretical paper (PRL 125, 037403 (2020)), the proposed structure, i.e., multiple graphene/dielectric layer stacks, is extremely challenging to fabricate. However, our proposed heterostructure and platform can be easily applied to other hyperbolic materials, e.g., MoO₃. In our demonstration, we experimentally showed that HPhPs can be changed by ~8 times with different doping concentration, which is significantly higher than the graphene case.

We have modified the introduction to highlight our advances:

Although HPhPs are volume-confined, they can still interact with the local environment through the evanescent field, and HPhP wavevectors are demonstrated to be tuned and engineered by changing the substrate permittivity in a list of studies¹⁻⁷, with this effect having been generalized by Fali. et al². On this track, one could use dynamic materials such as phase change materials⁸⁻¹⁰ and graphene¹¹⁻¹⁶ to actively modulate the HPhPs supported in the heterostructure.

And:

However, all existing demonstrations have either used 3D structures to induce a large contrast in the wavevectors, e.g., silicon versus air (etched silicon⁴³), or provided fundamentally limited polaritonic wavevector change (~1.6 times for phase change materials^{33,42} and ~ 1.2-2 times for graphene/hyperbolic media³⁵⁻⁴⁰).

(2) In those applications mentioned in the introduction part of current manuscript, many of them are requiring high-quality polaritons. The authors may want to show and discuss quality factor of the polariton modes as the substrate property is changing. In case the losses are high in the experiment, the authors may want to reargue the potential applications of their current platform.

We appreciate the reviewer's comment. We added a discussion of the HPhP loss and the potential applications. In short, the HPhP loss peaks around the modal order transition. We also proposed an application where the high loss is not a limiting factor: we can have guided HPhPs in suspended hBN region surrounded by hBN/InAs (Fig. S11). The hBN/InAs system only provides a medium with low wavevector, the loss caused by the low quality HPhP in the hBN/InAs is no longer critical to performance, instead only serving to provide a refractive index contrast.

Here we discuss the loss of HPhPs as a function of InAs plasma frequency. In the calculation, we assumed a constant scattering rate of InAs and calculated the HPhP wavevectors and figure of merits (FOM). The FOM is defined as:

$$FOM = \frac{\text{real}(k_{HPhP})}{\text{imag}(k_{HPhP})}$$

While the wavevector contrast is maximized around the modal order transition, the FOM is compromised significantly (Fig. S10). Although the FOM is relatively low around the modal order transition, it is still mostly higher than the FOM of HPhPs supported by naturally abundant hBN and MoO₃ along the [100] axis (~20-30) allowing for HPhP applications where manipulation of the polariton response would prove critical.

Fig. S10. HPhP FOM in a hBN/InAs heterostructure with varying InAs plasma frequency.

We further discuss a potential route to utilize the ultralow wavevector HPhPs without being affected by the associated loss: by using the hBN/InAs region as a cladding area to confine HPhP energy densities in a high wavevector region, e.g., a HPhP waveguide on suspended hBN, as shown in **Fig. S11b**. For a working frequency of 1500 cm⁻¹, hBN thickness of 100 nm, and InAs plasma frequency at 1500 cm⁻¹, the HPhP FOM on hBN/InAs area is only ~3, and the loss is too high to be useful. The k_{HPhP} is even lower than suspended hBN, indicating that a suspended hBN can serve as a “core material” to guide and confine HPhPs surrounded by hBN/InAs, with the concept being discussed thoroughly in our previous work¹⁷. With our previously developed analytical mode solution¹⁷, we found that such a guided mode has a FOM of ~40, which is comparable to many HPhP systems. We followed the same simulation strategy¹⁷ and verified the guided HPhPs in a suspended region (**Fig. S11a**).

Fig. S11. a. Finite element simulation showing a guided mode in suspended hBN region (top view), with relatively long propagation length. The suspended hBN is surrounded by hBN/InAs. b. The schematic of the guided HPhP on suspended hBN to utilize the low wavevector HPhPs on hBN/InAs region.

(3) The authors may want to discuss the detailed method to determine the transient plasma frequency of the substrate material after optical pumping in the experiment.

We apologize that the method was not clearly referred to in the main text. It was included in the supplementary information in the original submitted version, and we now modified the main text to discuss the data process principles.

In the pump-probe setup, we approximate the InAs as uniformly pumped (valid for our system as discussed in SI, section 12), and the transient InAs plasma frequencies were fitted with nano-FTIR spectra with a finite dipole model, and more details can be found in SI, section 12.

This work might be publishable after my above concerns are properly addressed.

Reviewer #3 (Remarks to the Author):

Review for “Polariton design and modulation via van der Waals / doped semiconductor heterostructures” by Mingze He et al. NCOMM-434944

In the manuscript, the authors study the tunability and manipulation of hyperbolic phonon polaritons using hyperbolic material/doped semiconductor platform. The study explores such tuning by utilizing the plasma frequency of an InAs/CdO substrate for nearly continuous HPhP wavevector tuning. Also, the authors show a sharp modal order transition when the plasma frequency of the doped semiconductor passes through the transitional frequency. Last, using ultrafast pulses, they show active modulation at picosecond timescales by photo-injecting carriers into the InAs substrate, showcasing a dynamic wavevector change of approximately 20%.

The authors have performed thorough research on the influence of substrate doping on the tunability of HPhP, combining theoretical and experimental research activities. I find the manuscript nicely written and

scientifically interesting. Thus, in general, I can support publication in Nat. Comm. However, I think that the manuscript still lacks several important information and needs further clarifications, thus, I cannot recommend publication in its current appearance.

We appreciate the reviewer's suggestions and comments, and please find the changes below.

The issues are described below (non-prioritized):

- I find the ultrafast response of the system very interesting. Not many groups can perform time-resolved scanning near-field optical microscopy and retrieve transient nano-imaging capabilities. However, I feel that the presented results and the discussion are quite laconic. For example:
 - o The authors show the response only at a specific spatial position from the edge (position $\sim 0.35 \mu\text{m}$ from the hBN edge). Have they checked other locations? Is the temporal dynamics remain the same in each location?

We appreciate the reviewer's comments. We performed another measurement at a different location, and the data was added to SI, section 11.

Another measurement at a different location shows a similar outcome (Fig. S19).

Fig. S19. Another measurement similar to Fig.4b but at a different spatial location. Similar measurements compared to the data in Fig. 4b in the main text. The measurement was performed at different spatial location ($\sim 0.5 \mu\text{m}$) from the edge of the same hBN, and the modulation characteristics are very similar to the data in the main text: the HPhP frequency is increased within 1 ps and the lifetime is about 8 ps.

I guess there is a spatio-temporal dynamics, but it could be a loose coupling in this system. I think that at least another measurement in a different location or two will check/resolve this issue.

We believe that there are temporal dynamic effects in the system. However, we cannot verify the effect in our measurement because we cannot measure the converted frequency since our probe laser is broadband. We are currently planning on more measurements with other techniques to discuss it thoroughly, which will be a separate study.

o Also, for different sample geometries and boundaries, the reflection from the boundaries suppose to vary, and in some cases multiple reflections from edges can be accumulated. Can/should this influence the temporal dynamics? for sure it will cause interferences. I think some discussion is missing.

We appreciate the reviewer's suggestion. Prior those nano-FTIR measurements, we performed AFM and s-SNOM mapping to confirm that we were doing line-profile scanning that is perpendicular to a hBN physical edge, so that we do not get complicated interferences from multiple edges. For instance, in Figure R2a region 2, complicated interferences from multiple edges are accumulated, while region 1 only contains clean and planewave-like polariton propagation. All our experimental data were collected in areas like region-1 to avoid such complicated interference effects that confound analysis (Figure R2b).

[REDACTED]

Figure R2 [REDACTED]. a, a s-SNOM data from reference showing “clean” and “interfered” polaritons in different regions²². b, Our near-field data were collected close to straight edges to have clean data.

o From the ultrafast dynamic evolution viewpoint – the pump causes free electron evolution, which is observed by the probe in various wavelengths. Have the authors examined what is the influence on the pump's intensity?

With different pump intensities, the concentration of free electrons can be changed. With higher pump intensity, the modulation can be stronger. We could not quantitatively examine the influence of the pump intensity because of the methodology that we used for alignment: the pump beam is focused on to the tip with a parabolic mirror, and we optimized the beam pointing direction so that the modulated InAs reflection measured by the probe beam is maximized. Since the alignment (focal beam spot size) varies between measurements, the pump intensity varies even with identical laser power, and cannot be compared directly.

o Any information on the group velocity of the phonon polaritons via such measurements?

The group velocity can be estimated following the procedure published by Yoxall et al²⁹ and Zhang et al³⁰. In short, one can perform a Fourier transform on the interferometer mirror position (i.e., the time delay between reference probe beam and scattered probe beam) converting the results into the frequency domain, and crop part of the frequency for inverse Fourier transform to acquire the polariton propagation group velocity. It only relies on the interference of probe beams and does not involve the use of pump beam. We added a short discussion in the text:

The group velocity in this panel was calculated, while one could directly extract the group velocity following the procedures described in references^{29,30}.

- The authors mention that other hyperbolic materials, such as MoO₃, can be an alternative to hBN. Still, it seems to be that interesting research works on tunable phonon-polaritons with MoO₃ relevant to the manuscript were not mentioned:

- o Ruta, Francesco L., et al. "Surface plasmons induce topological transition in graphene/ α -MoO₃ heterostructures." *Nature Communications* 13.1 (2022): 3719.

- o Zhang, Qing, et al. "Hybridized hyperbolic surface phonon polaritons at α -MoO₃ and polar dielectric interfaces." *Nano Letters* 21.7 (2021): 3112-3119.

We appreciate the suggestions and those references have been added.

- Some technical clarifications:

- o The demodulation order for the s-SNOM measurement is not specified in the paper or the supplementary materials. Specifying the order of the demodulation will provide information on the degree of near-field vs. far-field response.

- o The measurements are done with AFM's tip modulation of 100nm (for wavelengths of ~ 6um to 10um). This condition can cause the addition far-field information in the measured data. It will be nice to see the approach curve of the tips to observe the amount of far-field contribution.

We apologize that the related information was not specified in the submitted version. We have modified the method section.

We extracted the second harmonic signal for the nano-FTIR data, while for s-SNOM we used the 3rd harmonic to attain more near-field information. The AFM tapping amplitude was ~100 nm pre-approach, while the tapping amplitude during measurement was ~60-80 nm. We further characterized the approach curve for tapping amplitude of 75 nm and 115 nm respectively, and the 3rd harmonic signals are both dominated by the near-field signal (Fig. S20).

Fig. S20. The approach curves for 2nd and 3rd harmonic signals for different tapping amplitude. In both cases, the 3rd harmonic signals are dominated by the near-field contribution (the signal exponentially decays if the tip-sample distance is over ~20 nm (t in the figure)).

- How repeatable are the measurements for different samples? The author reported the results in several thicknesses. Do they have two samples with the same thicknesses? How close are the measured results?

The substrate tuning effect is highly repeatable among different samples (as shown in Fig.2 and Fig. 3a in the main text). We do have two samples with near identical thickness, which are shown in Fig. 1b and c, where dispersions differences are caused by the differences in the InAs carrier concentration. In Fig. 1d, the thickness is slightly different (55 nm instead of 51 nm), yet the dispersion is significantly different from panel b and c due to the metallic InAs. Symbols in all figure panels are experimental data, and the agreements between the experimental data and calculations are great.

• Clarifications on Figures:

o Figure 1a – could be more informative. Very nice illustration, but laconic. More graphical information of the various parameters (wavevector, thickness, etc.) can be added. The transition in Fig 1. B and C are both with thicknesses of 51nm (see caption)? If yes, why those are with different dispersion curves.

We apologize that we did not convey the tuning mechanism more clearly. This set of comparison is to stress that the dispersion of HPhPs could be tuned by the InAs substrate carrier concentration with (nearly) identical hBN thickness. We have modified the caption to more explicitly state claim:

Fig. 1. Tuned HPhP dispersion of hBN/doped semiconductor heterostructure by controlling semiconductor ω_p . a, Schematic of the platform. For the same hBN, the HPhP wavelength changes as a function of the plasma frequency of the semiconductor. hBN is represented with multi-layer hexagonal structures, with HPhPs shown as waves over it. In this example, the InAs on the right side (magenta color) is metallic with sufficiently high carrier concentration, shrinking the HPhP wavelength. b-d, Tuned HPhP dispersions by InAs plasma frequency. The plasma frequencies are noted on the corresponding panels, and we selected nearly identical thicknesses of hBN (51 nm, 51 nm and 55 nm respectively) to minimize the thickness dependence. The contour plots and dashed curves are calculated by transfer matrix method (TMM) and Eq. (1), respectively, and the triangles are extracted from s-SNOM data.

The figure has also been updated:

o Figure 2 – a proper legend is missing. Most of the caption text is devoted to explaining the data. I think adding a legend with information on the different symbols will help. (metallic CdO - purple rectangles, etc.). Same comment for Figure 3a.

We appreciate the reviewer’s suggestions. We have revised figure 2 and 3a accordingly:

To conclude, I find the manuscript scientifically interesting and informative, yet I feel that it still lacks some information, thus I cannot recommend publication in its current appearance. With these points adequately addressed, the manuscript will likely merit being published in Nat. Comm.

Reference.

- 1 Folland, T. G. *et al.* Reconfigurable infrared hyperbolic metasurfaces using phase change materials. *Nature Communications* **9**, 4371, doi:10.1038/s41467-018-06858-y (2018).
- 2 Fali, A. *et al.* Refractive Index-Based Control of Hyperbolic Phonon-Polariton Propagation. *Nano Letters* **19**, 7725-7734 (2019).
- 3 Ambrosio, A. *et al.* Selective excitation and imaging of ultraslow phonon polaritons in thin hexagonal boron nitride crystals. *Light: Science & Applications* **7**, 1-9 (2018).
- 4 Dai, S. *et al.* Phase-Change Hyperbolic Heterostructures for Nanopolaritonics: A Case Study of hBN/VO₂. *Advanced Materials* **31**, 1900251, doi:10.1002/adma.201900251 (2019).
- 5 Dai, S. *et al.* Hyperbolic Phonon Polaritons in Suspended Hexagonal Boron Nitride. *Nano Letters* **19**, 1009-1014, doi:10.1021/acs.nanolett.8b04242 (2019).
- 6 Kim, K. S. *et al.* The Effect of Adjacent Materials on the Propagation of Phonon Polaritons in Hexagonal Boron Nitride. *The Journal of Physical Chemistry Letters* **8**, 2902-2908, doi:10.1021/acs.jpcclett.7b01048 (2017).
- 7 Shen, J. *et al.* Hyperbolic phonon polaritons with positive and negative phase velocities in suspended α -MoO₃. *Applied Physics Letters* **120**, 113101 (2022).
- 8 Folland, T. G. *et al.* Reconfigurable infrared hyperbolic metasurfaces using phase change materials. *Nature Communications* **9**, 4371, doi:10.1038/s41467-018-06858-y (2018).
- 9 Chaudhary, K. *et al.* Polariton nanophotonics using phase-change materials. *Nature communications* **10**, 1-6 (2019).
- 10 Dai, S. *et al.* Phase-Change Hyperbolic Heterostructures for Nanopolaritonics: A Case Study of hBN/VO₂. *Adv Mater*, 1900251 (2019).
- 11 Dai, S. *et al.* Graphene on hexagonal boron nitride as a tunable hyperbolic metamaterial. *Nature nanotechnology* **10**, 682-686 (2015).
- 12 Álvarez-Pérez, G. *et al.* Active tuning of highly anisotropic phonon polaritons in van der Waals crystal slabs by gated graphene. *Acs Photonics* **9**, 383-390 (2022).
- 13 Zeng, Y. *et al.* Tailoring topological transitions of anisotropic polaritons by interface engineering in biaxial crystals. *Nano Letters* **22**, 4260-4268 (2022).
- 14 Ruta, F. L. *et al.* Surface plasmons induce topological transition in graphene/ α -MoO₃ heterostructures. *Nature communications* **13**, 1-7 (2022).
- 15 Hu, H. *et al.* Gate-tunable negative refraction of mid-infrared polaritons. *Science* **379**, 558-561 (2023).
- 16 Hu, H. *et al.* Doping-driven topological polaritons in graphene/ α -MoO₃ heterostructures. *Nature Nanotechnology* **17**, 940-946 (2022).
- 17 He, M. *et al.* Guided Mid-IR and Near-IR Light within a Hybrid Hyperbolic-Material/Silicon Waveguide Heterostructure. *Adv Mater* **33**, 2004305 (2021).
- 18 Folland, T. G. *et al.* Reconfigurable infrared hyperbolic metasurfaces using phase change materials. *Nature communications* **9**, 1-7 (2018).
- 19 Dai, Z. *et al.* Edge-oriented and steerable hyperbolic polaritons in anisotropic van der Waals nanocavities. *Nature communications* **11**, 1-8 (2020).
- 20 Herzig Sheinfux, H. *et al.* Transverse Hypercrystals Formed by Periodically Modulated Phonon Polaritons. *ACS Nano* **17**, 7377-7383 (2023).
- 21 Caldwell, J. D. *et al.* Sub-diffractive volume-confined polaritons in the natural hyperbolic material hexagonal boron nitride. *Nature communications* **5**, 5221 (2014).

- 22 Dai, S. *et al.* Tunable phonon polaritons in atomically thin van der Waals crystals of boron nitride. *Science* **343**, 1125-1129 (2014).
- 23 Passler, N. C., Jeannin, M. & Paarmann, A. Layer-resolved absorption of light in arbitrarily anisotropic heterostructures. *Physical Review B* **101**, 165425 (2020).
- 24 Govyadinov, A. A. *et al.* Recovery of permittivity and depth from near-field data as a step toward infrared nanotomography. *Acs Nano* **8**, 6911-6921 (2014).
- 25 Lee, I.-H. *et al.* Image polaritons in boron nitride for extreme polariton confinement with low losses. *Nature communications* **11**, 1-8 (2020).
- 26 Taboada-Gutiérrez, J. *et al.* Broad spectral tuning of ultra-low-loss polaritons in a van der Waals crystal by intercalation. *Nature materials*, 1-5 (2020).
- 27 Alfaro-Mozaz, F. J. *et al.* Hyperspectral Nanoimaging of van der Waals Polaritonic Crystals. *Nano Letters* (2021).
- 28 Hu, H. *et al.* Doping-driven topological polaritons in graphene/ α -MoO₃ heterostructures. *Nature Nanotechnology* **17**, 940-946, doi:10.1038/s41565-022-01185-2 (2022).
- 29 Yoxall, E. *et al.* Direct observation of ultraslow hyperbolic polariton propagation with negative phase velocity. *Nat Photonics* **9**, 674-678 (2015).
- 30 Zhang, X. *et al.* Ultrafast anisotropic dynamics of hyperbolic nanolight pulse propagation. *Sci Adv* **9**, eadi4407 (2023).

REVIEWERS' COMMENTS

Reviewer #1 (Remarks to the Author):

Most of my concerns are addressed. I would like to recommend its acceptance now.

Reviewer #2 (Remarks to the Author):

I think the paper can be published as it is.

Reviewer #3 (Remarks to the Author):

After carefully considering the authors' response to the reviewers, I must acknowledge that they have addressed the majority of my comments and the concerns raised during the review process. It is evident that they have made significant efforts to improve the quality and rigor of their research. However, while their responses are indeed extensive, it's essential to highlight that the most intriguing aspect of the study in my opinion, namely the exploration of temporal dynamics and spatio-temporal differences, still appears to be only partially examined.

I recommend that the authors explicitly mention this area as a valuable avenue for future research in the conclusion section of the paper. This addition would not only enhance the paper's academic value but also provide a valuable guidepost for researchers interested in expanding on this research in the future. Given the substantial improvements, I recommend this paper for publication in Nature Communications.

Response letter

REVIEWER COMMENTS

Reviewer #1 (Remarks to the Author):

Most of my concerns are addressed. I would like to recommend its acceptance now.

Reviewer #2 (Remarks to the Author):

I think the paper can be published as it is.

Reviewer #3 (Remarks to the Author):

After carefully considering the authors' response to the reviewers, I must acknowledge that they have addressed the majority of my comments and the concerns raised during the review process. It is evident that they have made significant efforts to improve the quality and rigor of their research. However, while their responses are indeed extensive, it's essential to highlight that the most intriguing aspect of the study in my opinion, namely the exploration of temporal dynamics and spatio-temporal differences, still appears to be only partially examined.

We appreciate the suggestion. We have added a comment in the discussion:

Importantly, the rise time of the system (below 1 ps) is relatively short compared with the group velocity of HPhPs, which could potentially lead to time-varying effects as demonstrated in radio-frequencies⁶¹. Therefore, it could potentially be a platform to explore spatial-temporal effects in the real space, yet significant efforts are needed.

I recommend that the authors explicitly mention this area as a valuable avenue for future research in the conclusion section of the paper. This addition would not only enhance the paper's academic value but also provide a valuable guidepost for researchers interested in expanding on this research in the future. Given the substantial improvements, I recommend this paper for publication in Nature Communications.